# Protocol for a randomized crossover study of thigh cuff inflation in experimental hemorrhage: Assessing its potential as a model for zone 3 REBOA

Sara Stadskleiv Torbjørnsen[1,2]*, Sole Lindvåg Lie[3], Marius Rehn[1,3,4], Jonny Hisdal[1,2], Lars Øivind Høiseth[1,5]

1 Faculty of Medicine, Institute of Clinical Medicine, University of Oslo, Oslo, Norway, 2 Section of Vascular Investigations, Heart-, lung- and vascular clinic, Oslo University Hospital, Oslo, Norway, 3 Department of Research and Development, Norwegian Air Ambulance Foundation, Oslo, Norway, 4 Air Ambulance Department, Division of Prehospital Services, Oslo University Hospital, Oslo, Norway, 5 Department of Anesthesia and Intensive Care Medicine, Division of Emergencies and Critical Care, Oslo University Hospital, Oslo, Norway

* saramsta@uio.no

## Abstract

Resuscitative endovascular balloon occlusion of the aorta (REBOA) is a method to provide temporary control of noncompressible torso hemorrhage in trauma patients. Previous research on REBOA has mainly focused on animals and patients. This study aims to explore whether thigh cuff inflation combined with simulated hemorrhage can serve as an experimental human model for zone 3 REBOA. Lower body negative pressure is a model of hypovolemia. A zone 3 REBOA occludes aorta at its bifurcation, essentially excluding the pelvis and lower extremities from the circulation. Bilateral proximal thigh cuffs will occlude blood vessels to and from the lower extremities. Twenty healthy volunteers will be exposed to bilateral proximal thigh cuff inflation to suprasystolic pressures to simulate the hemodynamic effects of REBOA during experimental hemorrhage using lower body negative pressure (LBNP). Each participant will complete two experimental conditions in a randomized order within one study visit. In the one condition, subjects will undergo only 60 mmHg LBNP for six minutes. In the alternate condition, 60 mmHg LBNP will be applied for six minutes, adding thigh cuff inflation during the final three minutes. Continuous, non-invasive monitoring of systemic hemodynamic parameters—including arterial blood pressure, stroke volume, and heart rate—will be conducted. Cerebral hemodynamics will be assessed by measuring middle cerebral artery blood velocity and cerebral oxygenation. Pain related to thigh cuff inflation will be assessed using a verbal numerical rating scale. The impact of thigh cuff inflation on systemic and cerebral hemodynamics will be evaluated using mixed-effects regression modeling. This study aims to examine the systemic and cerebral hemodynamic effects of combined thigh cuff inflation

**Data availability statement:** No datasets were generated or analysed during the current study. All relevant data from this study will be made available upon study completion. Additional data will be available from the authors for researchers who meet the criteria for access to confidential data by the Data Protection Officer, Oslo Norway (Contact via personvern@ous-hf.no).

**Funding:** The study was supported by The Research Council of Norway through the Medical Student Research Program at the University of Oslo and departmental resources. The funders had no role in study design, data collection and analysis, decision to publish, or preparation of the manuscript.

**Competing interests:** The authors have no competing interests.

and lower body negative pressure in healthy volunteers. Based on the feasibility and findings, the potential of this combination as a model for zone 3 REBOA in simulated hemorrhage will be discussed.

## Introduction

Among patients under the age of 45 years, trauma is the leading cause of death [1], with the majority of deaths occurring in the prehospital phase [2]. In both civilian and military settings, noncompressible torso hemorrhage (NCTH) represents the majority of potentially preventable trauma deaths [3,4]. Further treatment options are needed to improve the pre- and in-hospital survival of patients with NCTH.

Resuscitative endovascular balloon occlusion of the aorta (REBOA) is a method used in NCTH and hemorrhagic shock as a temporary measure before definitive hemorrhagic control can be achieved [5]. In REBOA, a balloon is inflated in the descending aorta, occluding blood flow and preventing further hemorrhage distal to the occlusion. The balloon can be positioned in three different zones in the descending aorta, depending on the location of the bleeding. Occlusion distal to the renal arteries is designated as zone 3 [6]. Since the introduction of REBOA in the 1960s, the method has more recently also been applied in the prehospital setting [7,8].

REBOA has profound hemodynamic effects [9]. Cardiac afterload and arterial blood pressure increase, potentially elevating cerebral perfusion pressure [10,11]. In addition to systemic hemodynamic effects, it is relevant to investigate cerebral hemodynamic effects, as polytrauma patients may suffer from traumatic brain injury (TBI). Using REBOA in these situations might increase the risk of intracranial hemorrhage and elevate intracranial pressure, potentially worsening patient outcomes. [12,13]. Although REBOA has demonstrated significant hemodynamic effects in animal and surgical settings, there remains a need for non-invasive, human experimental models to understand its physiological impact in awake individuals [14].

Lower body negative pressure (LBNP) is a model used to simulate hemorrhage in awake, healthy volunteers. Negative pressure is applied to the lower body, causing pooling of venous blood in the lower extremities and pelvis. Consequently, venous return and cardiac stroke volume are reduced, similar to what is observed during hemorrhage [15]. In contrast, arterial occlusion of the lower limbs has been shown to temporarily increase MAP, HR and systemic vascular resistance (SVR) [16]. Thigh cuff inflation achieves this arterial occlusion by restricting blood flow to the lower extremities and may therefore replicate some of the hemodynamic effects associated with zone 3 REBOA [17]. One previous study reported that a higher LBNP intensity was required to elicit similar reductions in stroke volume when arterial occlusion was combined with LBNP, compared to LBNP alone [18]. However, in that study, arterial occlusion was induced using tourniquets applied before, rather than during, LBNP.

Whether combining thigh cuff inflation and LBNP is a feasible method to investigate the hemodynamic effects of zone 3 REBOA during hemorrhage in healthy volunteers remains to be studied. The aim of this study is to investigate the systemic

and cerebral hemodynamic effects of thigh cuff inflation during LBNP in healthy volunteers. The primary hypothesis is that thigh cuff inflation will affect cardiac stroke volume during LBNP. The secondary hypothesis is that thigh cuff inflation will affect the response in mean arterial blood pressure (MAP) and middle cerebral artery velocity (MCAV) during LBNP.

## Materials and methods

### Study design

This is an ongoing, single-center, experimental study with a randomized, controlled crossover design. Subject recruitment began on April 4, 2024, and is anticipated to finish by February 28, 2025, with results expected by December 1, 2025. Each subject will undergo two experimental conditions in a randomized order during a single visit. In the one condition, 60 mmHg of LBNP will be applied for six minutes. In the other condition, 60 mmHg of LBNP will be applied for six minutes, with thigh cuff inflation added during the final three minutes [19]. A washout period of at least 20 minutes between the two conditions will be applied to minimize potential carry-over effects. Furthermore, any bias introduced by a possible residual carry-over effect is minimized by the randomized study design.

The experimental protocol is illustrated in **Fig 1**. Before the experiment begins, the subject will rest in the supine position for at least 20 minutes to stabilize hemodynamic parameters. Manual arterial blood pressure will be measured using a brachial cuff, and heart rate will be measured by ECG to define the LBNP stop criteria. After five minutes without LBNP, a 60 mmHg LBNP will be applied for six minutes, either with or without the addition of inflated thigh cuffs for the final three minutes. Following 20 minutes of supine rest, another five minutes of registration without LBNP followed by six minutes of 60 mmHg LBNP will be applied. During the second LBNP exposure, the opposite thigh cuff intervention will be used, ensuring that all subjects are exposed to LBNP both with and without inflated thigh cuffs (crossover).

LBNP will be terminated after completing six minutes of 60 mmHg, or earlier if the subject exhibits symptoms of impending hemodynamic decompensation such as blurred vision, lightheadedness, dizziness, nausea or sweating. Objective stop criteria for early termination include a reduction in heart rate or MAP to less than 75% of baseline that lasts more

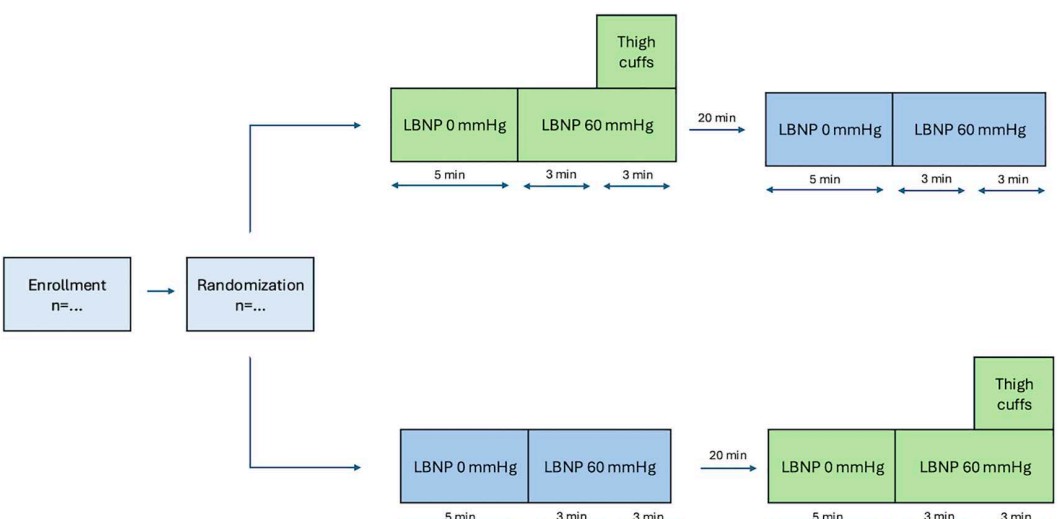

**Fig 1. Illustration of the experimental protocol.** The subject will participate in two experimental conditions. In randomized order, the subject will participate in one condition including lower body negative pressure (LBNP) 60 mmHg for six minutes and another experimental condition with both LBNP 60 mmHg for six minutes and thigh cuff inflation for the last three minutes.

than three seconds. Additionally, LBNP will be terminated at the subject's request for any reason other than those mentioned above.

## Subjects

Healthy volunteers will be included based on the eligibility criteria listed in **Table 1**. The subjects must refrain from caffeine (six hours), alcohol (24 hours) and strenuous exercise (on the test day) prior to the visit. They may eat a light meal up until two hours before the visit. A thorough introduction to the experimental setup will be provided before the subject gives written and oral consent to participate in the study. Each subject will receive a universal gift card of 500 NOK (approximately €43) to compensate for the inconvenience and discomfort of participating in the study.

## Randomization

The subjects will be randomized in a ratio of 1:1 to begin the test session with one of the two experimental conditions: LBNP alone or LBNP combined with thigh cuff inflation. To ensure an approximately equal number of subjects beginning the test session with either experimental condition, we will use block randomization with blocks of four or eight using the "blockrand" package in R [20]. Allocation will be concealed using sealed, opaque envelopes with sequential allocations, prepared in advance by a person otherwise not involved in the study. This will ensure that group assignment will remain unknown to the enrolling investigator until start of the experiment.

## Interventions

**Lower body negative pressure.** The subjects will be positioned in the supine position with deflated thigh cuffs. A saddle will be placed between the legs to minimize the use of leg muscles and prevent activation of the muscle-vein pump. The chamber will be air-sealed at the level of the iliac crest. The room temperature will be maintained between 22 and 24 °C. The model is illustrated in **Fig 2**.

**Thigh cuffs.** To approximate the hemodynamic effects of zone 3 REBOA, bilateral thigh cuffs will be positioned proximally on the thighs and inflated to a suprasystolic pressure during LBNP, as described in **Fig 2**. The thigh cuffs will be inflated to 80 mmHg above systolic blood pressure, as indicated by the manual arterial blood pressure measured at rest before LBNP. To minimize discomfort, we will use contoured thigh cuffs (CC22, 24x122.5cm cuff size Hokanson, Bellevue, WA, United States) with careful attention to finding the best and most comfortable placement as proximally as possible on the thighs. The thigh cuffs will be inflated rapidly using a Rapid Cuff Inflation device connected to Venopulse (STR Teknikk, Ålesund, Norway). Arterial occlusion will be confirmed by the disappearance of photoplethysmographic pulsation from a pulse oximeter placed on the first toe of the left foot.

## Measurements and data processing

Ascending aortic blood velocity will be measured using suprasternal pulsed Doppler ultrasound, with a 2 MHz probe (SD-50; Vingmed Ultrasound, Horten, Norway). Stroke volume will be calculated as the product of the aortic blood velocity-time

**Table 1. Eligibility criteria for the study population.**

- Age between 18 and 50 years
- No illnesses or health conditions that cause functional limitations in daily life or that require regular medication (except contraception or medication for allergies)
- Normal physical performance (e.g., able to walk and climb stairs without discomfort)
- No unexplained fainting episodes (previously assumed vasovagal syncope with a presumed known trigger, such as blood sampling, is acceptable)
- Not pregnant

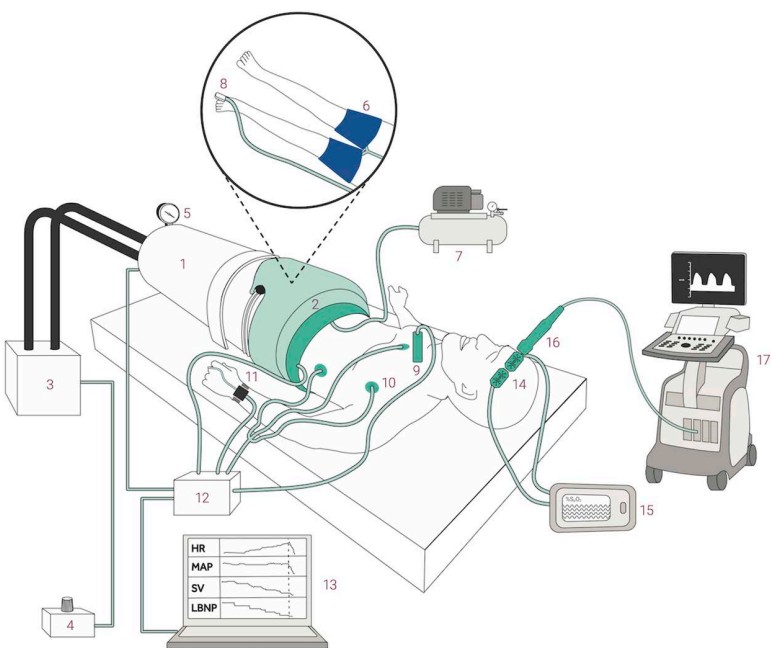

**Fig 2. Illustration showing the subject with thigh cuffs inside the LBNP chamber.** 1) The lower body negative pressure (LBNP) chamber is 2) sealed just above the iliac crest and connected to 3) a vacuum pump controlled by 4) a pressure control unit. The applied negative pressure is displayed on 5) a pressure monitor. 6) Thigh cuffs are positioned proximal on both thighs, inflated by 7) Venopulse. 8) The pulse oximeter is positioned at the first toe on the left foot, ensuring complete occlusion to the lower limbs. Measuring 9) ascending aortic blood velocity by suprasternal pulsed Doppler ultrasound, 10) ECG, 11) mean arterial pressure (MAP) on 12) a data acquisition device, 13) sampled continuously on a laptop. 14) Cerebral tissue oxygen saturation (ScO2) will be measured by near-infrared spectroscopy sampled on 15) an Invos monitor. 16) Mean cerebral artery velocity is sampled by a 1-5 MHz M5S probe connected to 17) Triplex transcranial Doppler ultrasound.

integral and the left ventricular outflow tract (LVOT) area. The diameter of the LVOT will be measured with echocardiography in the parasternal long-axis (1–5 MHz M5S probe, Vivid E95; GE Vingmed, Horten, Norway) [21], and the area will be calculated assuming a circular LVOT. The heart rate will be calculated from a three-lead ECG (Bio Amp/PowerLab, AD Instruments, Bella Vista, Australia). Cardiac output will be calculated as the product of heart rate and stroke volume.

The arterial blood pressure waveform will be measured continuously and noninvasively using the volume clamp method on the third finger of the left hand (Human NIBP Nano System; Finapres Medical System B.V., Netherlands). End-tidal $CO_2$ ($ETCO_2$) will be measured with sidestream capnography (Medlab CAP 10; Medlab GmbH, Stutensee, Germany). The photoplethysmography (PPG) waveform from the first toe on the left foot will be obtained using a pulse oximeter (Masimo Radical 7; Masimo Corp, CA, USA).

We will measure MCAV using triplex transcranial Doppler (TCD) ultrasound with a 1–5 MHz M5S probe (Vivid E95). The probe will be positioned over the right anterior temporal window and the middle cerebral artery (MCA) will be identified by its characteristic shape, depth (25–50 mm), and blood flow direction [22]. The sample volume will be placed to obtain a clear Doppler signal with a minimal insonation angle. The probe position and ultrasound machine settings, including the sample volume and angle will not change during both experimental conditions within each subject. The same operator will perform all TCD measurements to ensure reliability and uniformity in data collection. Cerebral tissue oxygen saturation ($ScO_2$) will be obtained using near-infrared spectroscopy (Invos 5100C cerebral/somatic oximeter; Somanetics, Troy, MI, USA).

Pain will be registered using a verbal numerical rating scale (VNRS), ranging from 0 (no pain) to 10 (worst imaginable pain). To minimize psychomotor and biofeedback responses during the protocol, subjects will be asked to rate their pain

once after completing both conditions. As pain was not a predefined endpoint, we will conduct a post hoc analysis of these data.

Stroke volume, heart rate, cardiac output, MAP, ETCO$_2$ and PPG waveform will be sampled at 1000 Hz using Lab Chart 8.1.28 (ADInstruments, Bella Vista, Australia). MCAV recordings will be exported as AVI-files to Cardiovascular Suite 4.0 (Quipu, Pisa, Italy) for Doppler waveform analysis to calculate the time-averaged maximum velocity. To synchronize the timing of the TCD recordings with the other recorded signals, the recordings will be mirrored directly into Lab Chart 8.1.28 via Digital Visual Interface (DVI) using a video grabber. ScO$_2$ data will be exported as.txt-files. Analyses will be performed in R [23] using RStudio [24].

## Outcomes

The primary outcome is the effect of thigh cuff inflation on stroke volume during LBNP. The secondary outcomes are the effects of thigh cuff inflation on MAP and MCAV during LBNP.

## Statistical methods

**Sample size.** We have previously observed a mean (standard deviation) reduction in stroke volume of 44 (11) % during 60 mmHg compared to zero mmHg of LBNP [25]. Based on these assumptions, simulations in R have shown that a sample size of 16 subjects will detect a 10% difference between the experimental conditions with α = 0.05 and (1-β)=0.90. To account for potential dropouts, we plan to enroll 20 subjects.

**Statistical analyses.** Due to the repetitive nature of the data and the clustering within subjects, the predefined primary and secondary hypotheses will be evaluated using mixed regression models, with subjects as a random effect. The details in the models (e.g., covariance structures and the use of generalized additive mixed models) will depend on the residual distribution and linearity of the data.

## Organization and conduct

This trial is approved by the Regional Ethics Committee (REK South-East D, ref. nr: 522416) and the Data Protection Officer at Oslo University Hospital (ref.nr: 22/22307). Sensitive data will be stored in a de-identified format on a secure data server at Oslo University Hospital. The linkage file will be stored in a separate software (Medinsight; Oslo University Hospital, Oslo, Norway). Data as "minimal data sets" will be available as Supporting Information upon publication of the original article. To ensure confidentiality, these data will not include demographic details. Further access to data may be granted upon reasonable request and approval by the Data Protection Officer.

# Discussion

In this present study, we will combine two established experimental interventions: LBNP and thigh cuff inflation. We thereby aim to explore whether this combination can serve as a model in awake humans to elucidate the systemic and cerebral hemodynamic effects observed with zone 3 REBOA in hemorrhage.

The hemodynamic response to REBOA in non-cardiac arrest settings has primarily been described in animal models. In a normovolemic canine model, ligation of the descending aorta led to increased systolic blood pressure and end-systolic and end-diastolic volumes of the left ventricle. Simultaneously, flow in the superior vena cava increased, indicating an increased flow to this vascular bed [26]

In a porcine model of hemorrhagic shock, REBOA increased proximal pressure, carotid artery blood flow, and brain oxygenation [27]. The combined effects of TBI and REBOA were studied in a rodent model. The setup and results were quite complex, but REBOA increased MAP and cerebral blood flow in the group with sham-TBI [28]. Although one pig model of TBI and hemorrhage did not find increased TBI-progression [29], the question of possible adverse effects of supranormal precerebral arterial pressures has been raised.

In a registry study, systolic blood pressure >160 mmHg was associated with worse outcomes in TBI-patients [30], and a case report described fatal intracerebral hemorrhage in a patient treated with REBOA for a pelvic hemorrhage who had a concomitant TBI [31]. However, the effects on MAP were less for zone 3 REBOA compared to zone 1 in a pig model with hemorrhage [32]. Therefore, conducting a study on the systemic and cerebral hemodynamic effects of hemorrhage and increased afterload in awake humans seems justified.

## Methodological considerations

To our knowledge, this is the first attempt to establish an experimental model of REBOA by combining LBNP and thigh cuffs. Accordingly, we have chosen to focus primarily on the main hemodynamic parameters. Depending on the results of the current study, future studies may incorporate laboratory analyses, such as arterial blood gas and lactate measurements, to further improve the model.

To the best of our knowledge, only one previous study has combined LBNP with arterial occlusion at thigh level [18]. However, in that study, lower limb tourniquets were applied before exposure to LBNP, potentially limiting the redistribution of blood to the lower limbs below the tourniquets. In contrast to REBOA, which only occludes the arteries, our model, similar to the tourniquet model, also occludes the veins. As venous congestion is already present in the lower extremities with LBNP, the consequences of additional venous occlusion due to thigh cuff inflation may be insignificant. This question remains to be elucidated.

Thigh cuff inflation will only occlude the vascular beds distal to the femoral arteries, thereby missing the pelvis in comparison to a zone 3 REBOA. One may consider occluding additional vascular beds, for example, by manually compressing the abdominal aorta against the columna. However, at this point, we do not wish to impose excessive hemodynamic interventions due to a lack of experience in combining these two interventions, as each in isolation includes quite profound hemodynamic effects.

While LBNP is generally not very uncomfortable, the application of thigh cuff inflation may cause pain, triggering the sympathetic nervous system and elevating the arterial blood pressure [25]. This may confound the hemodynamic effects of vascular occlusion. However, since both arterial occlusion and increased sympathetic activity tend to increase blood pressure, pain will not counteract the intended effect of vascular occlusion. To minimize pain and discomfort, we will use contoured thigh cuffs that are designed for a better fit and more even pressure distribution. In surgical settings, tourniquet cuffs are typically inflated to 100 mmHg above systolic blood pressure for arterial occlusion. Initially, we followed this standard, but after pilot testing, we reduced the cuff pressure. Using pulse oximetry, we confirmed the loss of distal pulsations as an objective indicator of successful occlusion and found that 80 mmHg above systolic consistently eliminated pulsatile signals while causing less discomfort than 100 mmHg.

Due to the pain and physical location, it is not feasible to blind either the subjects or the investigators to the thigh cuffs. An occlusion lasting a period of three minutes is within the safe inflation time of 1.5 to 2 hours [17].

LBNP is an established model of hypovolemia that has been used since the 1960s. It is considered to entail minimal risk in healthy humans. No serious adverse effects, such as deep venous thrombosis, have been reported [15]. Our research group has extensive experience utilizing LBNP to investigate various clinical scenarios [33,34], having included more than 100 subjects in the last 10 years.

It is important to note that there are differences between traumatic and simulated hemorrhage, as traumatic hemorrhage also involves tissue damage that leads to complex genomic, cellular, tissue and organ alterations [35]. In this present study, the subjects will be exposed to simulated hemorrhage for six minutes in each experimental condition, which is shorter than most clinical hemorrhagic cases. In addition, we will only study circulation in the healthy human brain, whereas the traumatized brain may behave differently.

We will use suprasternal Doppler ultrasound to obtain stroke volume [21]. It has been shown that the systolic velocity profile in the LVOT remains rectangular with little change in velocity from the aortic orifice for 3–4 cm above the aortic

valve, even if the aortic diameter changes slightly [21]. Thus, the sampling volume may be positioned in the ascending aorta to minimize noise and improve the quality of the Doppler signal without reducing the accuracy [21]. The ultrasound probe will be positioned in the suprasternal notch, a defined anatomical landmark. This localization will provide a fixed angle of insonation during the experiment [36], which is important as the measured velocity depends on the cosine of this angle. When measuring blood velocity in the MCA using TCD, the same considerations for the angle of insonation apply. It is therefore important not to move the ultrasound probe during the experiment. As the diameter of the MCA cannot be measured, it is assumed to be a constant throughout the experiment [22,37].

For safety reasons, LBNP and thigh occlusion will not be released simultaneously. Thigh occlusion will be relieved 30 seconds after LBNP to prevent excessive hypotension from relieving thigh occlusion before the effects of LBNP have subsided.

In recent years, partial REBOA has gained interest due to its association with lower rates of mortality and organ dysfunction compared to complete REBOA [38]. The technique involves controlled, partial inflation of an intra-aortic balloon catheter. This reduces distal blood flow while maintaining limited perfusion to organs distal to the occlusion site [39,40]. Complete REBOA is associated with a higher risk of complications in patients with hemorrhagic shock [41]. We may use insights from the present study to develop future models of partial occlusion, aiming to replicate the physiological effects of partial REBOA.

In conclusion, this present study aims to combine LBNP and thigh cuff inflation to investigate the systemic and cerebral hemodynamic effects of thigh cuff inflation during LBNP in healthy volunteers.

## Trial status

Data collection began in April 2024 and is expected to be completed in February 2025. The results will be submitted for publication in a peer-reviewed international journal and presented at scientific conferences.

## Author contributions

**Conceptualization:** Sara Stadskleiv Torbjørnsen, Sole Lindvåg Lie, Marius Rehn, Jonny Hisdal, Lars Øivind Høiseth.

**Data curation:** Sara Stadskleiv Torbjørnsen, Sole Lindvåg Lie.

**Formal analysis:** Lars Øivind Høiseth.

**Investigation:** Sole Lindvåg Lie, Lars Øivind Høiseth.

**Methodology:** Sara Stadskleiv Torbjørnsen, Sole Lindvåg Lie, Marius Rehn, Jonny Hisdal, Lars Øivind Høiseth.

**Project administration:** Sara Stadskleiv Torbjørnsen, Sole Lindvåg Lie, Lars Øivind Høiseth.

**Resources:** Lars Øivind Høiseth.

**Supervision:** Sole Lindvåg Lie, Marius Rehn, Jonny Hisdal, Lars Øivind Høiseth.

**Validation:** Sole Lindvåg Lie, Lars Øivind Høiseth.

**Visualization:** Sole Lindvåg Lie.

**Writing – original draft:** Sara Stadskleiv Torbjørnsen.

**Writing – review & editing:** Sole Lindvåg Lie, Marius Rehn, Jonny Hisdal, Lars Øivind Høiseth.

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
