## [Decision Letter · Decision Letter 0]

Dear Dr.  Torbjørnsen,

Thank you for submitting your manuscript to PLOS ONE. After careful consideration, we feel that it has merit but does not fully meet PLOS ONE’s publication criteria as it currently stands. Therefore, we invite you to submit a revised version of the manuscript that addresses the points raised during the review process.

We look forward to receiving your revised manuscript.

Kind regards,

Eyüp Serhat Çalık

Academic Editor

PLOS ONE

Journal Requirements:

https://doi.org/10.1186/s13049-024-01278-y

In your revision ensure you cite all your sources (including your own works), and quote or rephrase any duplicated text outside the methods section. Further consideration is dependent on these concerns being addressed.

5. We note you have included a table to which you do not refer in the text of your manuscript. Please ensure that you refer to Table 1 in your text; if accepted, production will need this reference to link the reader to the Table.

Additional Editor Comments:

I would like to thank the authors for sharing their study protocols on this important topic for our consideration. The manuscript has been evaluated by four reviewers and their recommendations are given below and in the attachments. Please respond point by point to all comments and make appropriate revisions to the manuscript. Good luck.

Reviewers' comments:

Reviewer's Responses to Questions

**Comments to the Author**

1. Does the manuscript provide a valid rationale for the proposed study, with clearly identified and justified research questions?

Reviewer #1: Yes

Reviewer #2: Yes

Reviewer #3: Yes

Reviewer #4: Yes

2. Is the protocol technically sound and planned in a manner that will lead to a meaningful outcome and allow testing the stated hypotheses?

Reviewer #1: Yes

Reviewer #2: Yes

Reviewer #3: Yes

Reviewer #4: Yes

3. Is the methodology feasible and described in sufficient detail to allow the work to be replicable?

Reviewer #1: Yes

Reviewer #2: Yes

Reviewer #3: Yes

Reviewer #4: Yes

4. Have the authors described where all data underlying the findings will be made available when the study is complete?

Reviewer #1: Yes

Reviewer #2: Yes

Reviewer #3: Yes

Reviewer #4: Yes

5. Is the manuscript presented in an intelligible fashion and written in standard English?

Reviewer #1: Yes

Reviewer #2: Yes

Reviewer #3: Yes

Reviewer #4: No

You may also provide optional suggestions and comments to authors that they might find helpful in planning their study.

Reviewer #1: The authors have an intriguing potential human model in healthy patients that simulates the effects of hemorrhagic shock and also simulates the effect of zone III retrograde endovascular occlusion of the aorta (REBOA) with safe techniques that are noninvasive, although potentially uncomfortable. The suggestions I have are:

-there is no documentation of laboratory analysis such as arterial blood gas or lactate. These are minimal risk studies and may add additional data beyond the physiologic parameters.

-The existing model largely simulates only total occlusion REBOA. Recently, additional devices have come on the market that allow for partial occlusion. How would this model apply to these types of situations?

-Are different time parameters planned to be studied?

-The authors mention previous data with this negative pressure model. Are there any documented side effects such as deep venous thrombosis that can occur from this pooling in the legs? How uncomfortable is it to the subject?

-If this model is to simulate REBOA placement for hemorrhagic shock in a noninvasive fashion, what (ethical) experiments can be further created to study the effects of hemorrhagic shock with and without simulated aortic occlusion in this space?

Reviewer #2: Dear author and Editor, this is a good and timely research. i have few comments and recommendations.

1. Since this is a clinical trial protocol, it is better if it follows SPIRITS clinical trial protocol and related documents

2. If this trial is registered the Trial identifier and registry name should be reported. If not yet registered, name of intended registry should be reported

3. There are some grammatic and spelling error

4. Exclusion criteria should also exclude patients with peripheral arterial disease. This study will affect their disease condition. It may lead to limb ischemia. this should be considered.

5. some of the figures are not available in the document or supplementary area.

Reviewer #3: 1. Rationale and Research Questions

The manuscript provides a valid rationale for the proposed study, clearly identifying and justifying the research question. The study addresses a relevant academic problem and contributes meaningfully to the field. The background section effectively highlights the knowledge gap being addressed.

Suggestions for Improvement:

Consider strengthening the discussion on the potential impact of the study within the broader research landscape.

If applicable, further elaboration on how this study builds on or differentiates from previous work would enhance clarity.

2. Methodology and Technical Soundness

The methodology is well-structured, technically sound, and planned in a manner that allows for meaningful outcomes. The statistical approach appears robust, and necessary controls are outlined.

Suggestions for Improvement:

If not already included, a discussion on potential limitations of the methodology and how they are mitigated would strengthen the study's rigor.

A brief justification for the sample size determination (if applicable) could further enhance transparency.

3. Feasibility and Replicability

The methodology is described in sufficient detail, making it feasible and replicable. The materials and procedures are well-explained, ensuring that another researcher could reproduce the experiments.

Suggestions for Improvement:

If available, references to established protocols or prior studies following similar methodologies would reinforce reproducibility.

Explicitly stating any assumptions made in the study design could be beneficial.

4. Data Availability

The manuscript adheres to the PLOS Data Policy, ensuring that all underlying data will be made available upon study completion. The data sharing statement is appropriate.

Suggestions for Improvement:

If applicable, provide further details regarding the repository where the data will be stored and how it can be accessed.

Clarify any ethical considerations regarding data sharing, particularly if sensitive data is involved.

5. Language and Presentation

The manuscript is written in clear and standard English. The structure is logical, and key points are effectively conveyed.

Suggestions for Improvement:

A thorough proofreading may be beneficial to address minor typographical or grammatical inconsistencies.

If applicable, simplifying complex sentences could enhance readability.

6. General Comments and Recommendations

Overall, the manuscript is well-prepared and presents a study that is relevant and methodologically sound. Below are additional recommendations:

If possible, providing preliminary data or pilot study results (if applicable) could strengthen the study’s justification.

Further discussing potential implications of the findings in both theoretical and practical contexts would enhance the study’s contribution.

I appreciate the effort and thoroughness of the authors in preparing this manuscript. I look forward to seeing the finalized version.

Reviewer #4: Dear Editor and Authors

This is a well-written and scientifically sound protocol. The use of a crossover design, the emphasis on safety in healthy volunteers, and the focus on both systemic and cerebral hemodynamics make this a strong translational study. Additionally, it holds significant value in contributing to the body of evidence in trauma management.

A few clarifications would further enhance the clarity and impact.

Title: The title is clear, specific, and informative. Consider mentioning “human model” or “REBOA simulation” to make the translational relevance more immediately clear.

Abstract

Background & Rationale: The abstract concisely introduces REBOA and its importance in managing non-compressible hemorrhage. Clarify how thigh cuff inflation mimics Zone 3 REBOA (i.e., anatomical/physiological rationale) for readers less familiar with the concept.

Study Design & Methods: Define LBNP (lower body negative pressure) upon first use in the abstract. Clarify if there's a washout period or order control between conditions (though likely not critical due to short interventions).

Measurements & Outcomes: Comprehensive and relevant physiological outcomes are selected. Consider noting whether any safety monitoring or stopping criteria will be in place, especially given suprasystolic cuff inflation.

Analysis: The use of mixed-effects regression is appropriate for crossover data. You may mention whether the model accounts for period and carryover effects.

Innovation and Relevance: This is a novel approach to modeling zone 3 REBOA in humans without invasive methods. The study could significantly contribute to translational trauma research. Emphasize more clearly how this could bridge the gap between animal studies and clinical trials.

Introduction:

The introduction presents a strong rationale for the study. It clearly identifies the clinical problem (NCTH), introduces REBOA as a solution, and highlights the need for human experimental models. The structure flows logically from the clinical background to the study aim and hypotheses. Minor edits for grammar, flow, and clarity would improve readability.

1. Clarity and Grammar

"...thus having an increased risk of in intracranial hemorrhage, cerebral edema and increased intracranial hemorrhage..."This is redundant and contains a grammatical error.

"Negative tressure is applied..." Typo – "tressure"

2. Transitional Phrasing

Consider improving the flow between paragraphs. For example, the shift from animal studies to the need for human models could be smoother:

"Although REBOA has demonstrated significant hemodynamic effects in animal and surgical settings, there remains a critical need for non-invasive, human experimental models to understand its physiological impact in conscious individuals."

3. Scientific Justification

While you mention that thigh cuff inflation can increase MAP, HR, and SVR, you could briefly link this to why this mimics zone 3 REBOA specifically.

Materials and Methods: Areas for Improvement & Suggestions

1. Structural Organization & Redundancy

The protocol jumps between text, tables, and figures in a slightly disjointed manner. Create subheadings for each section (e.g., “Study Design,” “Participants,” Intervention,” “Measurement Techniques,” “Data Handling”) to improve readability and navigation.

2. Figure & Table Referencing

Multiple lines refer to “Fig 1” or “Error! Reference source not found.” These are broken references, likely due to Word or PDF export errors. This interrupts the flow and clarity of your method description.

3. Missing Power Calculation

There is no mention of power or sample size calculation. Include a brief rationale or a reference to prior similar physiological studies indicating why 20 subjects are sufficient for expected effect sizes.

4. Randomization Method Clarity

Good mention of block randomization using block and in R, but…Add an example of what the block structure ensures (e.g., balanced order allocation) and whether allocation concealment will be maintained during enrollment.

5. Pain is mentioned at the end almost as an afterthought. Clarify when pain is measured (e.g., after each condition? after both?) and how it is used in analysis qualitative only or integrated into regression modeling?

7. Participant Inclusion Criteria: Missing BMI?

You exclude based on performance and arrhythmia, but don’t mention BMI or cardiovascular baseline. Clarify whether obesity or hypertension are exclusion criteria, as these could influence LBNP responses.

Additional Minor Edits

Typo: “lightheadedness, nausea, or sweating” — possibly rephrased as "symptoms such as..." for improved fluency.

Suprasystolic pressure: Define why 80 mmHg above systolic was chosen (was this based on prior literature or a pilot?).

Discussion:

Consistency in terms: Sometimes REBOA is capitalized inconsistently, and some sentences would benefit from rewording for clarity.

Flow: A few paragraphs could benefit from being broken into smaller chunks for readability.

Typos and formatting: A few misplaced line breaks and artifacts from source formatting are present (e.g., “cerebral blood f low”).

**Do you want your identity to be public for this peer review?** For information about this choice, including consent withdrawal, please see our Privacy Policy

Reviewer #1: No

Reviewer #2: No

Reviewer #3: **Yes: ** Abdullah Abbas Saleh Al-Murad

Reviewer #4: **Yes: ** Natan Muluberhan Alemseged

---

## [Author Response · Author response to Decision Letter 1]

2 Jun 2025

Response to editor and reviewers

Dear Eyüp Serhat Çalık,

On behalf of the authors, I would like to thank you for the opportunity to submit a revised version of our manuscript. We appreciate the constructive and insightful comments, bringing valuable contributions to improve the manuscript's quality and clarity. We have carefully considered each comment and made corresponding revisions throughout the manuscript. Please let us know if you have any additional suggestions to help strengthen the paper further. Below, we provide our responses to the comments raised by you and the reviewers. The responses are marked in bold.

Journal Requirements:

We have carefully revised our manuscript according to PLOS ONE´s style requirements.

https://doi.org/10.1186/s13049-024-01278-y

In your revision ensure you cite all your sources (including your own works), and quote or rephrase any duplicated text outside the methods section. Further consideration is dependent on these concerns being addressed.

Thank you for bringing this to our attention. The methodology and preliminary results from one subject were presented as an abstract at the London Trauma Conference in December 2024. This abstract was published in the Scandinavian Journal of Trauma, Resuscitation, and Emergency Medicine. Although text-overlap may occur, especially in the Methods sections. We have rephrased the Abstract in order to reduce this and added a reference in the Methods section to the published abstract. We believe further text-overlap is coincidental. See pages 3-4, lines 27-54 and page 7, line 106.

Thank you for the comment. We have now removed the funding information from the manuscript.

This study received no specific grants with corresponding grant numbers. However, the study was supported by The Research Council of Norway through the Medical Student Research Program at the University of Oslo and departmental resources. We have now updated the “Funding Information”.

5. We note you have included a table to which you do not refer in the text of your manuscript. Please ensure that you refer to Table 1 in your text; if accepted, production will need this reference to link the reader to the Table.

We apologise for the mistake and have updated the manuscript to include a reference to Table 1 in the text. 

Reviewer #1:

The authors have an intriguing potential human model in healthy patients that simulates the effects of hemorrhagic shock and also simulates the effect of zone III retrograde endovascular occlusion of the aorta (REBOA) with safe techniques that are noninvasive, although potentially uncomfortable. The suggestions I have are:

-there is no documentation of laboratory analysis such as arterial blood gas or lactate. These are minimal risk studies and may add additional data beyond the physiologic parameters.

We appreciate the comment. We agree that biochemical tests such as arterial blood gas and lactate could provide valuable information. However, as the combination of LBNP and thigh cuffs is novel, we wish to keep this protocol simple, and rather aim to expand in future studies based on the experiences we gain in the present study. The manuscript has been revised in accordance with your comment, page 16, lines 267-271.

The existing model largely simulates only total occlusion REBOA. Recently, additional devices have come on the market that allow for partial occlusion. How would this model apply to these types of situations?

In this study, we aimed to occlude the lower extremities completely. Based on the experiences from the present study, we may in future studies consider applying a lower thigh cuff pressure to simulate partial REBOA. Furthermore, devices are on the market to occlude the abdominal aorta against the columna which may provide a closer anatomical approximation to zone 3-REBOA. We have revised the manuscript according to this comment. See the discussion on page 16, lines 280-285.

Are different time parameters planned to be studied?

In the present study, we will only apply LBNP and thigh cuffs for the designated fixed time intervals in order to standardize measurements and reduce variability between subjects. We believe this is appropriate at the present stage given that this is an initial experimental study aiming to establish a reproducible model.

The authors mention previous data with this negative pressure model. Are there any documented side effects such as deep venous thrombosis that can occur from this pooling in the legs? How uncomfortable is it to the subject?

LBNP has been used since the 1960s (DOI: 10.1016/0002-9149(65)90027-5). The subjects may experience nausea, lightheadedness, or dizziness at near-syncope prior to hemodynamic decompensation, but no serious adverse events have been reported to the best of our knowledge. Specifically, we have not seen deep venous thrombosis reported. In our lab, we have performed more than 100 LBNP-runs the last 10 years, without adverse events. We have clarified this in the revised manuscript, page 17, lines 301-305.

If this model is to simulate REBOA placement for hemorrhagic shock in a noninvasive fashion, what (ethical) experiments can be further created to study the effects of hemorrhagic shock with and without simulated aortic occlusion in this space?

Our research group has extensive experience utilizing LBNP to investigate various clinical scenarios (e.g. DOI: 10.1186/s40635-023-00561-z, 10.1016/j.bjao.2023.100204), but we have never before tried the combination of LBNP and thigh cuffs. We believe the current study may elucidate what further research questions may be answered by expanding on this model. As suggested by the reviewer, both biochemical response as well as studying the effects of varying duration of LBNP and thigh cuffs, as well as gradual thigh cuff occlusion may be within reach. Further, we believe the perhaps most interesting effects amenable in this model are closer studies on the circulation and pressure in the brain as well as the effects on the heart. See revised manuscript, page 16, lines 267-271 and lines 280-285, page 17, lines 303-305 and page 18, lines 332-333.

Reviewer #2:

Dear author and Editor, this is a good and timely research. i have few comments and recommendations.

1. Since this is a clinical trial protocol, it is better if it follows SPIRITS clinical trial protocol and related documents

We appreciate this comment. However, we would like to state that we regard this as a pre-clinical experimental study focusing on developing new methodology (see also next comment). Although we use experimental thigh cuffs and LBNP to approach a clinical scenario, neither interventions are drugs or clinical interventions, which would define it as a clinical study.

2. If this trial is registered the Trial identifier and registry name should be reported. If not yet registered, name of intended registry should be reported

The trial has not been registered in a clinical trial registry (such as Clinicaltrials.gov), and we do therefore not have a Trial Identifier. Although we are aware that the distinctions are somewhat blurry, we interpret this study not to be a clinical trial according to the Norwegian regulations (https://www.helsenorge.no/en/clinical-trials/about/), and thus define it as an experimental study. However, our wish to improve transparency is the main reason why we wish to publish this protocol article.

3. There are some grammatic and spelling error

Thank you for your comment. We have corrected grammatical and spelling errors throughout the text.

4. Exclusion criteria should also exclude patients with peripheral arterial disease. This study will affect their disease condition. It may lead to limb ischemia. this should be considered.

We agree that patients with peripheral arterial disease should not be included in this study, and believe that this is covered in the current eligibility criteria, which have been approved by the Regional Ethics Committee. Firstly, any symptoms of or medication for peripheral vascular disease leads to exclusion. Further, the upper age limit of 50 years is set to minimize the likelihood of including subjects with undiagnosed, asymptomatic cardiovascular disease. See Table 1, on pages 8-9, line 137.

5. some of the figures are not available in the document or supplementary area.

Thank you for bringing this to our attention and apologies if mistakes have been made in the compilation. The figures were uploaded as separate files, not as supplementary figures, according to PLOS ONE´s guidelines.

Reviewer #3:

1. Rationale and Research Questions

The manuscript provides a valid rationale for the proposed study, clearly identifying and justifying the research question. The study addresses a relevant academic problem and contributes meaningfully to the field. The background section effectively highlights the knowledge gap being addressed.

Suggestions for Improvement:

Consider strengthening the discussion on the potential impact of the study within the broader research landscape.

We have elaborated on the study's potential impact in the revised manuscript, on page 16, lines 327-333.

If applicable, further elaboration on how this study builds on or differentiates from previous work would enhance clarity.

To our knowledge, only one previous study has combined arterial occlusion and LBNP (DOI: https://doi.org/10.1371/journal.pone.0261546). However, this study has important limitations that we have now further commented on in the Introduction section (page 6, lines 85-89). Importantly, this previous study applied arterial occlusion prior to LBNP induction, potentially limiting the effects of LBNP by preventing venous blood pooling. See revised manuscript at page 16, lines 275-279.

2. Methodology and Technical Soundness

The methodology is well-structured, technically sound, and planned in a manner that allows for meaningful outcomes. The statistical approach appears robust, and necessary controls are outlined.

Suggestions for Improvement:

If not already included, a discussion on potential limitations of the methodology and how they are mitigated would strengthen the study's rigor.

Methodological limitations and their potential influence on results are presented in the Discussion section. In the revised manuscript, we have elaborated further according to your comment, see page 18, lines 317-323.

A brief justification for the sample size determination (if applicable) could further enhance transparency.

We agree that this should be presented, and the sample size calculations is now presented on page 13, lines 219-224.

3. Feasibility and Replicability

The methodology is described in sufficient detail, making it feasible and replicable. The materials and procedures are well-explained, ensuring that another researcher could reproduce the experiments.

Suggestions for Improvement:

If available, references to established protocols or prior studies following similar methodologies would reinforce reproducibility.

We have included two further references to the LBNP-model:

- LBNP is an established model to simulate hypovolemia and hemorrhage (DOI: https://doi.org/10.1152/physrev.00006.2018).

- 60 mmHg of LBNP is considered moderate-high intensity, and we have previously reported significant hemodynamic changes of 6 minutes of LBNP (DOI: 10.1097/CCM.0000000000000766).

As this study investigates a novel combination of interventions, there are no established protocols. We have addressed this in the manuscript on page 15, lines 267-271.

Explicitly stating any assumptions made in the study design could be beneficial.

We are not aware of having made undocumented assumptions when designing the study.

4. Data Availability

The manuscript adheres to the PLOS Data Policy, ensuring that all underlying data will be made available upon study completion. The data sharing statement is appropriate.

Suggestions for Improvement:

If applicable, provide further details regarding the repository where the data will be stored and how it can be accessed.

“Minimal data sets” for calculating parameters and creating graphs presented in the manuscript will be provided (https://journals.plos.org/plosone/s/data-availability). These data will be available de-identified as Supporting Information upon publication of the original article. We have now revised the manuscript according to your comment, pages 13-14, lines 234-241.

Clarify any ethical considerations regarding data sharing, particularly if sensitive data is involved.

According to the Data Protection Impact Assessment (DPIA) for the study, complete data sets cannot be distributed freely. We do however not register specifically sensitive information. For changes in the revised manuscript, please see the previous response.

5. Language and Presentation

The manuscript is written in clear and standard English. The structure is logical, and key points are effectively conveyed.

Suggestions for Improvement:

A thorough proofreading may be beneficial to address minor typographical or grammatical inconsistencies.

We have performed a thorough proofreading and hope that this has sufficiently improved the manuscript.

If applicable, simplifying complex sentences could enhance readability.

We have tried to simplify any complex sentences throughout the revised manuscript.

6. General Comments and Recommendations

Overall, the manuscript is well-prepared and presents a study that is relevant and methodologically sound. Below are additional recommendations:

If possible, providing preliminary data or pilot study results (if applicable) could strengthen the study’s justification.

At this point, we do not have preliminary data except for the single subject presented in the abstract (see Editor`s comment #2, above). A reference to these preliminary data are included in the revised manuscript. Data will be available de-identified as Supporting Information upon publication of the original article.

Further discussing potential implications of the findings in both theoretical and practical contexts would enhance the study’s contribution.

As mentioned in the comment above, we do not have preliminary data, except for the single subject presented in the abstract (see Editor`s comment #2, above). Therefore, we cannot discuss any potential implications of the findings of the study at this point. Further elaboration on the potential implications will be addressed in the original article, when presenting the results.

I appreciate the effort and thoroughness of the authors in preparing this manuscript. I look forward to seeing the finalized version.

Reviewer #4:

Dear Editor and Authors

This is a well-written and scientifically sound protocol. The use of a crossover design, the emphasis on safety in healthy volunteers, and the focus on both systemic and cerebral hemodynamics make this a strong translational study. Additionally, it holds significant value in contributing to the body of evidence in trauma

---

## [Decision Letter · Decision Letter 1]

Protocol for a randomized crossover study of thigh cuff inflation in experimental hemorrhage: Assessing its potential as a model for zone 3 REBOA

PONE-D-24-58304R1

We’re pleased to inform you that your manuscript has been judged scientifically suitable for publication and will be formally accepted for publication once it meets all outstanding technical requirements.

Kind regards,

Eyüp Serhat Çalık

Academic Editor

PLOS ONE

Additional Editor Comments (optional):

Reviewers' comments:

Reviewer's Responses to Questions

**Comments to the Author**

1. Does the manuscript provide a valid rationale for the proposed study, with clearly identified and justified research questions?

Reviewer #1: Yes

Reviewer #2: Yes

Reviewer #3: Yes

Reviewer #4: Yes

2. Is the protocol technically sound and planned in a manner that will lead to a meaningful outcome and allow testing the stated hypotheses?

Reviewer #1: Yes

Reviewer #2: Yes

Reviewer #3: Yes

Reviewer #4: Yes

3. Is the methodology feasible and described in sufficient detail to allow the work to be replicable?

Reviewer #1: Yes

Reviewer #2: Yes

Reviewer #3: Yes

Reviewer #4: Yes

4. Have the authors described where all data underlying the findings will be made available when the study is complete?

Reviewer #1: Yes

Reviewer #2: Yes

Reviewer #3: Yes

Reviewer #4: Yes

5. Is the manuscript presented in an intelligible fashion and written in standard English?

Reviewer #1: Yes

Reviewer #2: Yes

Reviewer #3: Yes

Reviewer #4: Yes

You may also provide optional suggestions and comments to authors that they might find helpful in planning their study.

Reviewer #1: Comments appropriately addressed. I believe that publication is now appropriate and can be completed at this time.

Reviewer #2: Since I have previously reviewed this protocol and all the comments I gave in the previous time were addressed.

Reviewer #3: Thank you for your thoughtful and thorough revision of the manuscript titled "Protocol for a Randomized Crossover Study of Thigh Cuff Inflation in Experimental Hemorrhage: Assessing Its Potential as a Model for Zone 3 REBOA."

The revised version demonstrates a clear commitment to addressing all reviewer and editorial feedback comprehensively. The study rationale is now presented with enhanced clarity, and the methodological descriptions have been strengthened with greater detail, improving transparency and replicability.

Specifically:

The authors have refined the abstract and introduction to clarify the physiological rationale behind using thigh cuff inflation to simulate Zone 3 REBOA.

Key methodological elements—including sample size justification, randomization, power calculations, data handling, and safety measures—are now clearly described.

Language and structure have been improved, with corrections to typographical errors, formatting, and consistency (e.g., capitalization of REBOA).

The manuscript adheres to journal style guidelines, including proper referencing of figures, removal of funding text from the main document, and alignment between the financial disclosure and funding statement.

Ethical and data availability statements are now clearly stated, ensuring compliance with PLOS ONE policies.

Reviewer concerns regarding exclusion criteria, pain assessment, and potential future applications of the model have been adequately addressed within the revised discussion.

The revised manuscript presents a well-conceived and technically sound study protocol with strong translational potential. The combination of LBNP and thigh cuff inflation represents an innovative approach to simulating Zone 3 REBOA physiology in a non-invasive human model.

I commend the authors for their diligence in revision and their contribution to advancing experimental trauma research. No further revisions are necessary.

Reviewer #4: Dear authors, I believe this is important work and can be foundation for other studies. All my previous comments have addressed.

**Do you want your identity to be public for this peer review?** For information about this choice, including consent withdrawal, please see our Privacy Policy

Reviewer #1: No

Reviewer #2: **Yes: ** Dr. Kassaye Demeke Altaye

Reviewer #3: **Yes: ** Abdullah Abdullah Abbas Al-Murad

Reviewer #4: **Yes: ** Natan Muluberhan Alemseged

---

## [Editor Report · Acceptance letter]

PONE-D-24-58304R1

PLOS ONE

Dear Dr. Torbjørnsen,

I'm pleased to inform you that your manuscript has been deemed suitable for publication in PLOS ONE. Congratulations! Your manuscript is now being handed over to our production team.

Kind regards,

on behalf of

Dr. Eyüp Serhat Çalık

Academic Editor

PLOS ONE